# SCGAN-generated skin-tone-balanced extension of the HAM10000 dataset for fairer dermatological AI

**Safrina Kabir** [ID]*

Department of Electrical and Electronic Engineering, Islamic University of Technology, Gazipur, Bangladesh

* safrinakabir@iut-dhaka.edu

## Abstract

This dataset presents a synthetic collection of skin lesion images generated using a tone-conditioned Generative Adversarial Network (GAN) to model skin tone variation in dermatological datasets, motivated by the limited representation of darker skin tones in commonly used benchmarks. The dataset includes 10,000 images, at a resolution of 128×128 pixels, equally divided between medium and dark skin tones, with lesion class labels assigned to follow the class distribution of the original HAM10000 dataset. To quantitatively characterize the generated data, we report Fréchet Inception Distance (FID) scores and a baseline classification experiment comparing models trained on HAM10000 alone versus HAM10000 augmented with the proposed synthetic images. This work provides a controlled synthetic extension intended to support fairness-aware experimentation, data augmentation, and downstream evaluation in dermatological machine learning.

## Introduction

Skin cancer detection using deep learning has seen significant progress in recent years, with datasets like HAM10000 becoming benchmarks for automated lesion classification [1]. However, a persistent challenge remains: skin tone imbalance in dermoscopic image datasets [2,3]. The overwhelming majority of publicly available lesion images are from light-skinned individuals, leading to models that are less well characterized and potentially less reliable when applied to darker skin tones [4]. This work introduces a synthetic extension of the HAM10000 dataset designed to explore skin tone variation through controlled image generation, rather than to directly correct dataset bias. Using a tone-conditioned Skin-Conditioned Generative Adversarial Network (SCGAN), we generate dermoscopic images conditioned on explicit skin tone vectors while preserving the original lesion class distribution [5,6]. Unlike traditional GANs, the proposed framework explicitly conditions generation on skin tone and lesion class, and applies color-guided blending to simulate pigmentation variation,

**Data availability statement:** The synthetic dataset and associated generation code are publicly hosted on IEEE DataPort (DOI: https://doi.org/10.21227/51cx-yy74). The entry includes: • 10,000 synthetic skin lesion images (5,000 medium tone and 5,000 dark tone) in PNG format at 128×128 resolution. • Metadata in CSV format (class labels following HAM10000 distribution). • README.TXT with dataset descriptions and usage instructions. • The custom SCGAN code (Python scripts implementing the tone-conditioned GAN architecture, training pipeline, and color blending post-processing) bundled in a zipped folder. The code is provided with necessary documentation to reproduce the dataset generation, including dependencies (e.g., PyTorch), hyperparameters, and execution steps.

**Funding:** The author(s) received no specific funding for this work.

**Competing interests:** The authors have declared that no competing interests exist.

while structure-preserving objectives are used to maintain lesion morphology [7,8]. The resulting dataset consists of 10,000 synthetic lesion images (5,000 medium tone and 5,000 dark tone), spanning all HAM10000 diagnostic categories, and is intended to support quantitative evaluation, data augmentation studies, and fairness-aware experimentation in dermatological machine learning [9].

## Materials and methods

A conditional GAN architecture was designed with tone conditioning using a two-dimensional one-hot encoded vector (medium: [1,0], dark: [0,1]). The generator leverages a ResNet-based design, and the discriminator is trained with adversarial loss, classification loss, and L1 reconstruction loss [10]. During training:

- Input noise vectors were sampled from a standard normal distribution.

- Tone conditions were randomly assigned to enforce diversity.

- Color blending was applied post-generation to simulate realistic skin tone ranges.

- Training was conducted on the HAM10000 dataset at 128×128 resolution for 100 epochs [11].

The generator was trained to produce tone-consistent lesion images, while discriminator training ensured realism and tone classification accuracy. The output images were saved with blending applied to ensure earthy tone realism (40 percent for medium, 80 percent for dark) [12]. The size 128×128 pixels was selected to align with common input dimensions used in lightweight convolutional neural networks, especially those designed for mobile or memory-constrained diagnostic tools. While higher resolutions may capture finer morphological details, 128×128 offers a practical balance between computational efficiency and visual fidelity for training and validating deep learning models, particularly in fairness-aware dermatological AI applications. To match the original HAM10000 class distribution, lesion labels were allocated to synthetic images proportionally based on the original dataset's metadata, ensuring diversity across lesion types. Fig 1 shows the overall methodology of training and generation of data.

### Data acquisition and preprocessing

This dataset is built on the HAM10000 dataset, which consists of 10,015 dermoscopic images classified into seven categories: Actinic keratosis (akiec), Basal cell carcinoma (bcc), Benign keratosis-like lesions (bkl), Dermatofibroma (df), Melanoma (mel), Melanocytic nevi (nv), and Vascular lesions (vasc). An overview of the HAM10000 dataset is shown in Fig 2.

To initialize the synthesis pipeline, a subset of lesion images was selected from HAM10000 and organized into a folder-based class hierarchy compatible with torchvision.datasets.ImageFolder. The images were subjected to a consistent preprocessing pipeline to ensure uniformity in model inputs. Each image was resized to a resolution of 128×128 pixels, converted into PyTorch tensors, and normalized to the range [−1, 1] using a mean and standard deviation of 0.5. Class labels were retained

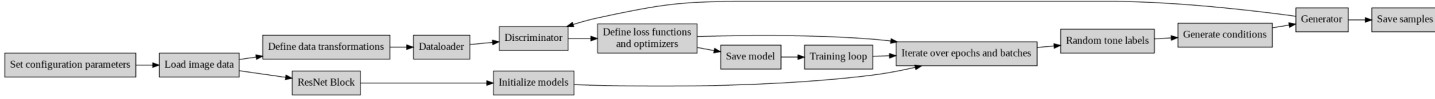

**Fig 1. Overall methodology flowchart.**

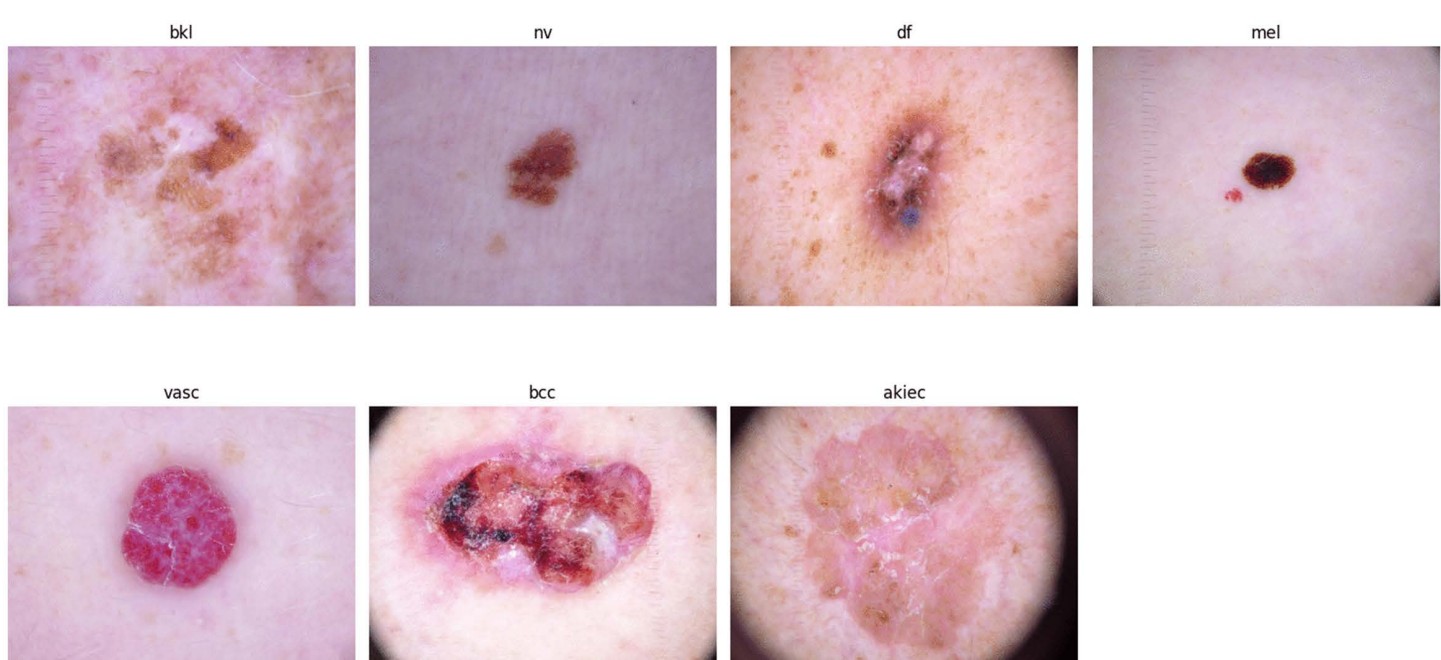

**Fig 2. Samples from HAM10000 dataset.**

from the original metadata and later used to proportionally assign diagnostic labels to the generated synthetic images, preserving the original class distribution.

## Conditional generator and discriminator design

The generator receives as input a spatial noise tensor sampled from a standard normal distribution, concatenated channel-wise with a broadcast skin tone condition vector. The generator architecture comprises:

- An initial convolutional block with 7×7 kernels

- Two downsampling layers that double feature dimensionality

- Six residual blocks to preserve lesion structure while enabling tone variation

- Two upsampling layers using transposed convolutions

- A final Tanh activation to normalize outputs to the range [−1, 1] The discriminator is implemented as a dual-headed convolutional network tasked with:

- Adversarial discrimination (real vs. synthetic)

- Tone classification (medium vs. dark)

The discriminator input consists of an image concatenated with its corresponding tone condition vector, broadcast across spatial dimensions. A PatchGAN-style output is used to improve sensitivity to local texture and structural consistency.

## Optimization strategy & color-tint synthesis

Three loss functions were employed to balance realism, tone specificity, and structural preservation:

- Adversarial loss (MSELoss) to promote visual realism

- Classification loss (CrossEntropyLoss) to enforce tone conditioning

- Reconstruction loss (L1Loss) to encourage structural similarity between real and generated images The generator and discriminator were optimized using the Adam optimizer with a learning rate of 0.0002, $\beta_1 = 0.5$, $\beta_2 = 0.999$. During training, tone labels were randomly assigned within each batch to encourage diversity and prevent mode collapse. The models were trained at a resolution of 128×128 pixels, which was selected to balance computational efficiency and representational capacity, particularly for lightweight or resource-constrained diagnostic applications. To enhance tone realism, post-generation color blending was applied to the synthesized images. Fixed RGB tint values were defined for each tone:

- Medium: [0.682, 0.439, 0.227] with 40 percent blending

- Dark: [0.384, 0.227, 0.094] with 80 percent blending

The tint values were broadcast across image tensors and linearly blended with generator outputs to simulate a range of skin pigmentation while preserving lesion morphology. The blending process for medium and dark tones is illustrated in Fig 3 and Fig 4, respectively.

## Baseline Classification Experiment

To quantitatively assess the utility of the synthetic dataset, a baseline classification experiment was designed using a standard convolutional neural network. A ResNet-18 architecture pretrained on ImageNet was selected due to its widespread use and computational efficiency. Two training configurations were evaluated:

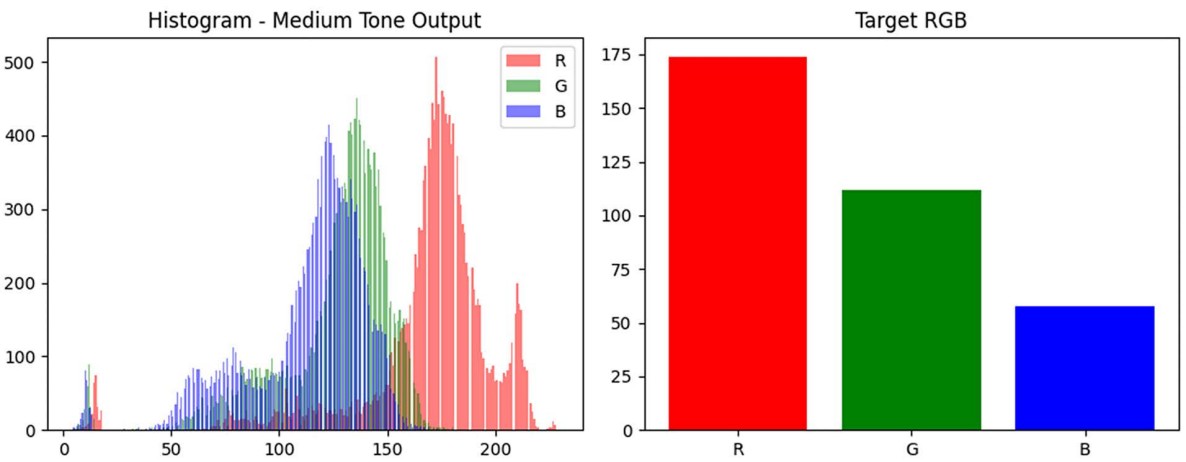

**Fig 3. RGB tint blending for medium tone.**

**Fig 4. RGB tint blending for dark tone.**

1. A baseline model trained solely on the original HAM10000 dataset

2. A model trained on HAM10000 augmented with the proposed synthetic images

All images were resized to 128×128 pixels to maintain consistency with the generation resolution. The models were trained using identical optimization settings, loss functions, and evaluation protocols to ensure a controlled comparison. Performance was evaluated using accuracy and macro-averaged F1-score to account for class imbalance.

## Results

The GAN successfully generated 5,000 medium-tone and 5,000 dark-tone skin lesion images with visible lesion textures, tone realism, and consistent resolution. Visual inspection confirms the distinguishability of tones, realism of blending, and variation across lesion morphology. These qualitative observations suggest that the proposed conditioning and color blending strategy produces visually coherent samples across diagnostic categories. Fig 5 and Fig 6 show the comparison of samples of an epoch for two different skin tones.

Equal number of samples are generated for medium tone and dark tone under each category of lesion classes. Number of generated samples per class in listed in Table 1 and distribution of samples across the classes is illustrated in Fig 7.

To characterize the generated dataset, Fréchet Inception Distance (FID) was computed between the synthetic images and the original HAM10000 images resized to 128×128 pixels. The resulting FID scores indicate a measurable domain gap, which is expected given the resolution reduction and explicit tone manipulation applied during generation. These scores are reported as a reference measure of visual similarity rather than as an indicator of clinical realism.

Fréchet Inception Distance (FID) scores:

• FID (HAM10000 + all synthetic): 141.935

• FID (HAM10000 + synthetic medium): 141.9928

• FID (HAM10000 + synthetic dark): 169.8825

The baseline classification experiment demonstrated improved performance compared to a model trained on HAM10000 alone, as measured by overall accuracy and macro-averaged F1-score. This improvement, shown in Fig 8,

Medium Tone - Epoch 96

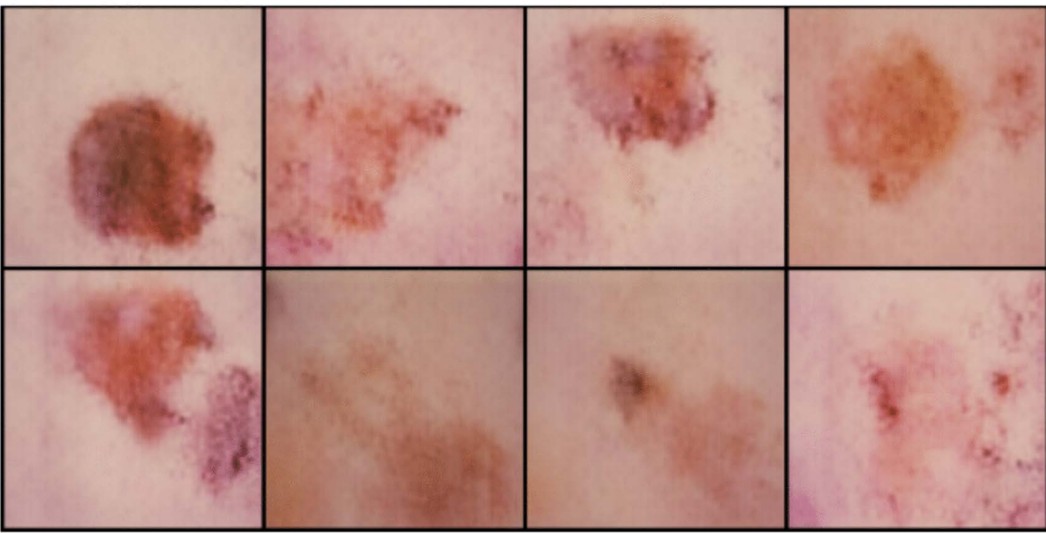

**Fig 5. Medium skin tone generated samples.**

Dark Tone - Epoch 96

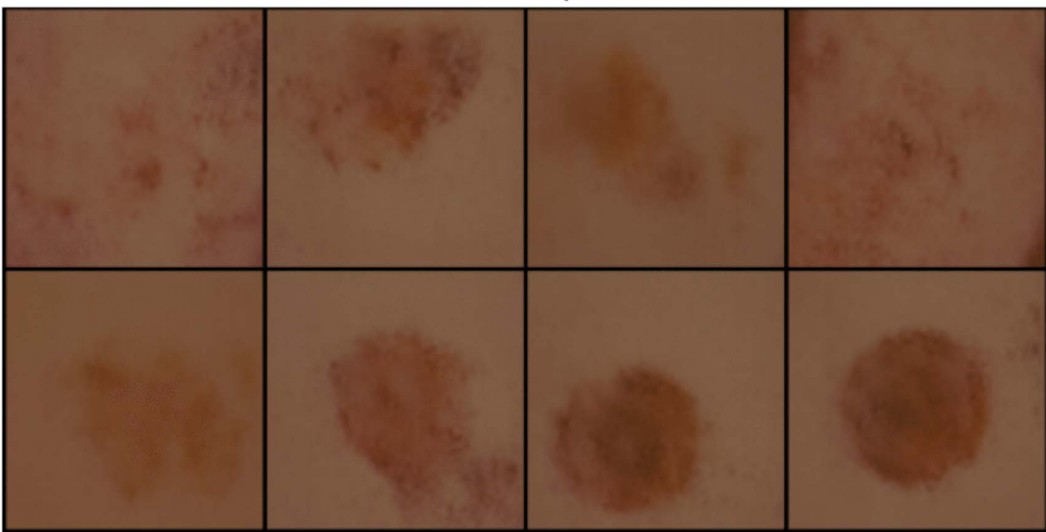

**Fig 6. Dark skin tone generated samples.**

suggests that the synthetic images contribute useful supervisory signals for downstream learning, despite the modest absolute performance levels.

Baseline classification results:

- HAM10000 only: Acc = 0.1774, Macro-F1 = 0.1625

- HAM10000 + synthetic: Acc = 0.2486, Macro-F1 = 0.2403

**Table 1. Synthetic Image Allocation per Class.**

| Class | Medium | Dark |
|---|---|---|
| akiec | 163 | 163 |
| bcc | 257 | 257 |
| bkl | 549 | 549 |
| df | 57 | 57 |
| mel | 556 | 556 |
| nv | 3347 | 3347 |
| vasc | 71 | 71 |

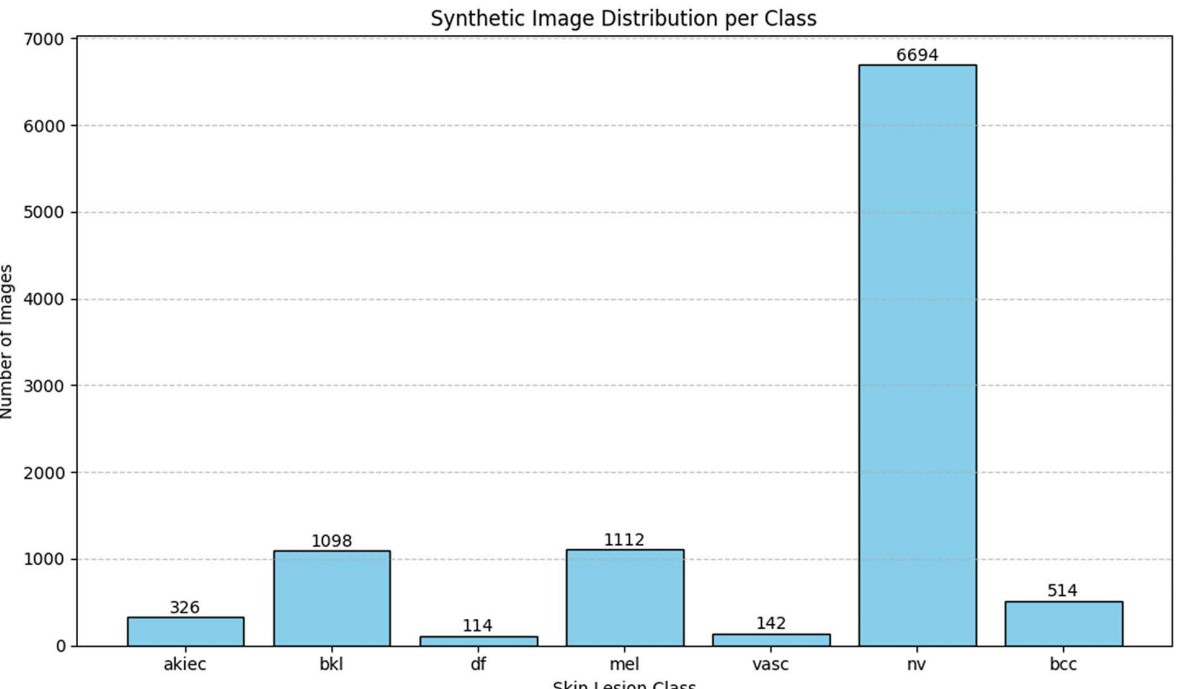

**Fig 7. Synthetic images generated Across the classes.**

These quantitative results are intended to provide empirical context for the dataset's potential utility, rather than to establish state-of-the-art classification performance.

## Discussion

This work presents a synthetic extension of the HAM10000 dataset generated using a tone-conditioned generative framework. The primary goal of the study was not to claim clinical realism or diagnostic superiority, but to provide a controlled dataset that enables systematic exploration of skin tone variation in dermatological machine learning tasks.

### Interpretation of quantitative results

The Fréchet Inception Distance (FID) scores reported in this study indicate a noticeable distribution gap between the generated images and the original HAM10000 dataset. This outcome is expected for several reasons. First, all images were

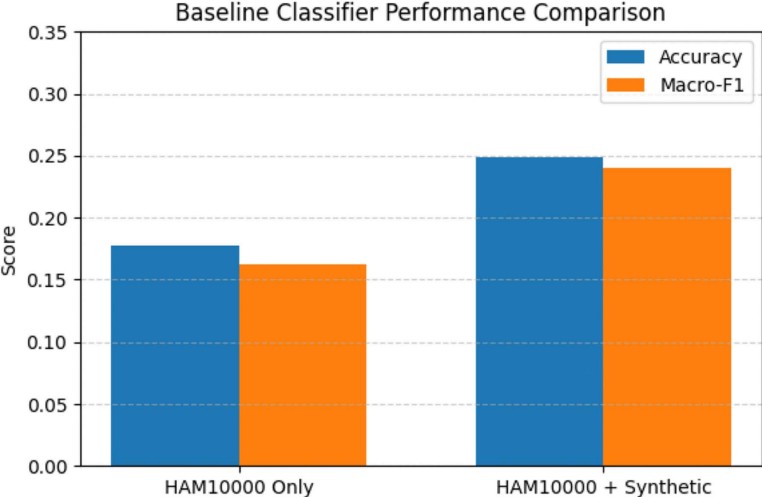

**Fig 8. Baseline Classification.**

resized to 128×128 pixels to ensure computational feasibility and architectural consistency, which inevitably reduces fine-grained morphological detail. Second, explicit tone manipulation and post-generation color blending introduce controlled deviations from the original image distribution. As a result, the FID values should be interpreted as reference indicators of visual similarity rather than as measures of diagnostic realism. The baseline classification experiment provides complementary quantitative evidence. Although the absolute classification accuracy and macro-averaged F1-score remain modest, the consistent improvement observed when training with the augmented dataset suggests that the synthetic images contribute informative supervisory signals. This improvement is particularly relevant in the context of class imbalance and limited representation of non-light skin tones, where additional diversity—even if imperfect—can improve feature learning. Importantly, this experiment was designed as a reference-level evaluation rather than as an optimized classification benchmark.

### Why absolute performance remains modest

Several factors contribute to the relatively low absolute performance of the baseline classifier. The reduced input resolution limits the availability of high-frequency texture cues commonly used in dermatological diagnosis. In addition, HAM10000 exhibits significant class imbalance, particularly in minority classes such as dermatofibroma and vascular lesions, which adversely affects macro-averaged metrics. Finally, the baseline classifier was intentionally kept simple to avoid confounding architectural optimization with the effects of dataset augmentation. These design choices reflect the study's focus on dataset characterization rather than model performance maximization.

### Justification for using SCGAN

Generating skin lesion images with controllable skin tones requires a GAN that balances realism, lesion feature preservation, and tone control [13]. Standard GANs (vanilla GAN, DCGAN) can generate realistic images but cannot explicitly control attributes like skin tone [14]. Conditional GANs allow label-based control but may struggle to maintain fine lesion details across tone variations [15]. Multi-domain frameworks such as Smart GAN, StarGAN and CycleGAN excel at translating existing images between domains but are less suited for large-scale synthetic generation, especially when target skin tones are underrepresented or inconsistently labeled [16]. SCGAN overcomes these limitations by injecting

tone-conditioning vectors into both the generator and discriminator, ensuring explicit control over skin tone while preserving lesion morphology. It allows stable, large-scale synthesis without relying on source images from underrepresented domains, making it ideal for fairness-driven dataset augmentation.

## Implications and limitations

Although the images were generated using a tone-conditioned GAN with perceptual and edge-preserving losses, they are still synthetic and may not fully capture the complex biological variations found in real-world skin lesions [4]. The model was trained using light-toned images as reference. As a result, it may not reflect nuances seen in real medium- or dark-skinned lesions across varied populations [17,18]. Since the original HAM10000 dataset lacks skin tone labels and is inherently biased toward lighter tones, the conditional SCGAN is limited by the representational capacity of its source [19].

While this dataset provides a lightweight and accessible synthetic extension of HAM10000 with a focus on skin tone balance, all images were generated at a resolution of 128×128 pixels to support memory-efficient machine learning workflows [20]. In future iterations, we plan to generate and release higher-resolution versions (e.g., 512×512 pixels) to enable broader use cases such as fine-grained feature analysis, dermoscopic segmentation, and clinical decision support. These future releases will also incorporate more diverse lighting conditions, lesion types, and metadata fields to expand the dataset's utility.

In summary, this study demonstrates that tone-conditioned generative modeling can be used to construct a controlled synthetic extension of a widely used dermatological dataset. Through qualitative inspection and reference-level quantitative evaluation, the proposed dataset is shown to be suitable for experimentation in data augmentation, fairness-aware analysis, and generative benchmarking. By explicitly documenting both its capabilities and limitations, this work aims to support transparent and reproducible research in dermatological artificial intelligence.

## Value of the data

- **Supports research in AI fairness for dermatology:** The original HAM10000 dataset is heavily skewed toward fair-skinned individuals, which can lead to bias in automated skin lesion classification models [21]. This dataset provides a controlled synthetic extension focusing on medium and dark skin tones, enabling researchers to study representation gaps and conduct fairness-aware analyses in dermatological machine learning.

- **Enables training/testing of classifiers on underrepresented skin tones:** By introducing 10,000 synthetic images across medium and dark tones—balanced across all 7 diagnostic classes, this dataset allows machine learning models to be evaluated under expanded skin tone conditions, particularly for melanin-rich skin tones that are underrepresented in commonly used benchmarks.

- **Useful for benchmarking skin tone robustness in existing models:** Researchers can use this dataset to analyze whether skin lesion classifiers exhibit performance variation across skin tones under controlled conditions. It enables comparison of model accuracy, confidence calibration, and error behavior between lighter and darker skin images under identical lesion class conditions.

- **Encourages further research on tone-aware synthetic augmentation using conditional GANs:** This dataset demonstrates the successful use of a Skin-tone Conditional GAN (SCGAN) to generate realistic, tone-specific lesion images. It can serve as a benchmark and inspiration for future work on data-driven bias mitigation, fairness-aware generative modeling, and tone-adaptive medical image synthesis.

- **Supports education and awareness around bias in clinical AI workflows:** The dataset can also be used in academic settings to teach students, practitioners, and policymakers about the implications of bias in clinical imaging

datasets. It provides a practical resource for teaching dataset limitations, bias analysis, and controlled experimentation in medical AI workflows.

- **Open-source and reusable across domains:** Licensed under Creative Commons (CC BY 4.0), the dataset is freely available for research and education. Its format is simple (.png images organized by class and tone), making it easy to integrate into existing pipelines for training, validation, and interpretability studies.

## Acknowledgments

This dataset was developed as part of an independent AI fairness initiative. I thank the contributors of the HAM10000 dataset for making the source data publicly available.

## Author contributions

**Conceptualization:** Safrina Kabir.

**Data curation:** Safrina Kabir.

**Methodology:** Safrina Kabir.

**Software:** Safrina Kabir.

**Visualization:** Safrina Kabir.

**Writing – original draft:** Safrina Kabir.

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
