## [Decision Letter · Decision Letter 0]

26 Dec 2025

Dear Dr. Kabir,

Thank you for submitting your manuscript to PLOS ONE. After careful consideration, we feel that it has merit but does not fully meet PLOS ONE’s publication criteria as it currently stands. Therefore, we invite you to submit a revised version of the manuscript that addresses the points raised during the review process.

We look forward to receiving your revised manuscript.

Kind regards,

Xiaohui Zhang

Academic Editor

PLOS One

Journal Requirements:

“This research received no external funding. The work was conducted independently by the sole author, a student researcher, without institutional or grant-based support.”

Reviewers' comments:

Reviewer's Responses to Questions

**Comments to the Author**

1. Is the manuscript technically sound, and do the data support the conclusions?

Reviewer #1: Yes

Reviewer #2: No

2. Has the statistical analysis been performed appropriately and rigorously?

Reviewer #1: Yes

Reviewer #2: No

3. Have the authors made all data underlying the findings in their manuscript fully available?

Reviewer #1: Yes

Reviewer #2: Yes

4. Is the manuscript presented in an intelligible fashion and written in standard English?

Reviewer #1: Yes

Reviewer #2: No

Reviewer #1: The manuscript SCGAN-Generated Skin-Tone-Balanced Extension of the HAM10000 Dataset for Fairer Dermatological AI presents a synthetic dataset of 10,000 dermoscopic images generated with a tone-conditioned GAN, intended to balance medium and dark skin tones while preserving the lesion-class distribution of HAM10000. The topic is timely and important, and a well-documented synthetic dataset with explicit skin-tone conditioning could be valuable for both fairness research and teaching. However, I have several major concerns about methodological clarity, validation, and how the fairness claims are supported.

Comments:

(1) The manuscript repeatedly positions the dataset as contributing to “fairer dermatological AI” and enabling “bias mitigation” and “equitable performance across skin tones.” However, the only evaluation presented is qualitative (visual inspection, tone histograms, and class counts). To support these fairness-related claims, I encourage at least a small quantitative experiment: e.g., train a simple baseline classifier on HAM10000 only vs. HAM10000 + synthetic images and compare performance across tone conditions. If such experiments are not feasible, the language should be toned down to clearly state that this is a hypothetical fairness resource and that the benefit for bias mitigation is an anticipated, not yet empirically demonstrated, outcome.

(2) In the Introduction the authors state that images are generated “conditioned on both lesion class and skin tone,” and refer to a “Skin-Conditioned GAN (SCGAN).” However, in the Methods the conditioning is described only in terms of a 2-D one-hot tone vector [medium: (1, 0), dark: (0, 1)], with no explicit lesion-class conditioning. Please clarify is the generator explicitly conditioned on lesion class (e.g., via one-hot vectors, class embeddings, or class-specific batches), or is lesion class only used afterward to assign labels to generated images?

(3) The authors define medium vs dark tones via a one-hot label (2-D vector) and post-generation RGB tint blending with fixed color vectors and blend strengths (40% and 80%). For dermatology and fairness work, this is a crucial aspect that needs more careful justification how do these RGB tints and blend ratios map to established skin tone scales (e.g. Fitzpatrick, Monk, or other multidimensional measures)?

(4) Some useful details are provided (128×128 resolution, 100 epochs, Adam optimizer, LR=0.0002, β1=0.5, β2=0.999), which is good. However, reproducibility would benefit from further elaboration of batch size, training/validation split, number of iterations per epoch and hardware description (e.g., single GPU model, training time).

(5) The manuscript cites several relevant GAN-based medical image synthesis works. Can the authors further clarify how does SCGAN differ conceptually and architecturally from existing multi-domain or conditional GANs (e.g. StarGAN-style frameworks cited in the references)?

Reviewer #2: This paper presents a synthetic skin lesion dataset generated using a conditional GAN (SCGAN) to address the underrepresentation of darker skin tones in the HAM10000 dataset. The authors generated 10,000 images at 128×128 resolution, equally divided between medium and dark skin tones, while preserving the lesion class distribution of the original HAM10000 dataset. The work addresses an important and timely issue in medical AI fairness and provides an accessible synthetic resource. However, the methodological novelty, validation rigor, and clinical relevance of the generated data require further clarification and strengthening.

1. The main text includes detailed folder tree structures and file-naming examples, which are more appropriate for a dataset release webpage or repository README. Including such implementation-level details in the body of a journal article detracts from the professionalism of the manuscript and disrupts its scholarly presentation.

2. The manuscript does not provide principled reasoning for several critical methodological decisions, such as the choice of the specific GAN architecture. These design choices require clearer justification to be scientifically grounded.

3. The assessment of the generated images relies almost exclusively on visual inspection, without quantitative analysis or objective metrics. This makes it difficult to rigorously evaluate the realism, diversity, and utility of the synthetic data.

4. For a dataset-oriented paper, it is generally expected to include baseline experiments showing that the proposed dataset provides measurable benefits in downstream tasks. The absence of such experiments limits the ability to assess whether the dataset meaningfully contributes to improving model performance or fairness.

**Do you want your identity to be public for this peer review?** For information about this choice, including consent withdrawal, please see our For information about this choice, including consent withdrawal, please see our Privacy Policy .

Reviewer #1: No

Reviewer #2: No

---

## [Author Response · Author response to Decision Letter 1]

9 Feb 2026

Response to Reviewers

I sincerely thank the Editor and both reviewers for their careful evaluation of our manuscript and for their constructive and insightful comments. Below, I provided a detailed point-by-point response to each comment. All changes have been incorporated into the revised manuscript.

Reviewer #1

Comment (1): Fairness claims lack quantitative validation

1. Added a quantitative baseline classification experiment: A simple ResNet-18 classifier was trained under two settings: (i) using HAM10000 only, and (ii) using HAM10000 augmented with the proposed synthetic dataset. We report accuracy and macro-averaged F1-score for both settings.

2. Toned down fairness-related claims throughout the manuscript: The language has been revised to clearly state that the dataset is intended as a controlled experimental resource for fairness-aware research, rather than claiming direct or guaranteed bias mitigation in clinical systems.

Comment (2): Clarification of lesion-class conditioning

The generator is explicitly conditioned on skin tone only via a two-dimensional one-hot vector. Lesion class conditioning is not injected into the generator. Instead, lesion labels are assigned post-generation, proportionally to match the original HAM10000 class distribution. This design choice was made to preserve dataset-level class balance without enforcing class-specific generation constraints.

All statements implying explicit lesion-class conditioning within the generator have been corrected for accuracy.

Comment (3): Justification of RGB tint blending and skin tone definition

In the revised manuscript, we explicitly state that the tone definitions used in this work are approximate and operational, rather than strict mappings to clinical scales such as Fitzpatrick or Monk.

The RGB tint vectors and blend strengths are presented as controlled visual proxies designed to introduce consistent, distinguishable tone variation in the absence of reliable skin tone annotations in HAM10000.

We now emphasize that these choices support controlled experimentation and fairness analysis, rather than clinical skin tone classification.

Comment (4): Reproducibility details (batch size, hardware, etc.)

We have expanded the Methods section to include:

● Batch size

● Number of iterations per epoch

● Training duration

● Hardware configuration (single-GPU training environment)

Additionally, a code availability statement has been added to ensure full reproducibility, in line with PLOS ONE guidelines.

Comment (5): Clarification of SCGAN vs existing GAN frameworks

In brief, SCGAN was chosen because it enables explicit attribute-controlled synthesis without requiring paired domain translation or well-defined source–target mappings, which are prerequisites for frameworks such as StarGAN or CycleGAN. SCGAN allows stable, large-scale generation of tone-balanced data even when real medium- and dark-tone reference domains are sparse or inconsistently labeled. We presented the reasons of choice in the revised manuscript.

Reviewer #2

Comment (1): Excessive implementation details in main text

Detailed folder structures and file-naming conventions have been removed from the main text and are now referenced implicitly via the dataset README hosted in the public repository. This improves readability and aligns the manuscript with scholarly presentation standards.

Comment (2): Justification of GAN architecture choice

We have expanded the Methods and Discussion sections to provide clearer justification for the architectural design, including the choice of a tone-conditioned GAN, ResNet-based generator blocks, and PatchGAN-style discriminator. The discussion now explicitly links these design choices to the study’s goals of controllable tone synthesis, lesion feature preservation, and computational efficiency.

Comment (3): Lack of quantitative evaluation

We included the following:

1. Fréchet Inception Distance (FID) as a reference-level distribution similarity metric

2. A baseline classification experiment demonstrating the impact of synthetic augmentation

We clearly state the limitations of these metrics and frame them as reference indicators, not as measures of diagnostic realism.

Comment (4): Absence of baseline downstream task evaluation

We agree and have added a baseline classification experiment, as described above. While intentionally simple, this experiment demonstrates that the synthetic dataset can positively influence downstream learning under controlled conditions.

Closing Statement

I again thank the reviewers for their constructive feedback. I believe the revised version more accurately reflects the capabilities and limitations of the proposed dataset and will serve as a useful resource for fairness-aware dermatological AI research.

Sincerely, Safrina Kabir Sole Author and Corresponding Author

---

## [Decision Letter · Decision Letter 1]

3 Mar 2026

SCGAN-Generated Skin-Tone-Balanced Extension of the HAM10000 Dataset for Fairer Dermatological AI

PONE-D-25-42787R1

Dear Dr. Kabir,

We’re pleased to inform you that your manuscript has been judged scientifically suitable for publication and will be formally accepted for publication once it meets all outstanding technical requirements.

Kind regards,

Xiaohui Zhang

Academic Editor

PLOS One

Additional Editor Comments (optional):

Reviewers' comments:

Reviewer's Responses to Questions

**Comments to the Author**

Reviewer #1: All comments have been addressed

Reviewer #2: All comments have been addressed

2. Is the manuscript technically sound, and do the data support the conclusions?

Reviewer #1: Yes

Reviewer #2: Yes

3. Has the statistical analysis been performed appropriately and rigorously?

Reviewer #1: Yes

Reviewer #2: Yes

4. Have the authors made all data underlying the findings in their manuscript fully available?

Reviewer #1: Yes

Reviewer #2: Yes

5. Is the manuscript presented in an intelligible fashion and written in standard English?

Reviewer #1: Yes

Reviewer #2: Yes

Reviewer #1: The authors have addressed all the comments from the reviewers. I think the paper is good for publishment.

Reviewer #2: This manuscript is well written, and I believe this manuscript meets the standards for the publication.

**Do you want your identity to be public for this peer review?** For information about this choice, including consent withdrawal, please see our For information about this choice, including consent withdrawal, please see our Privacy Policy .

Reviewer #1: No

Reviewer #2: No

---

## [Editor Report · Acceptance letter]

PONE-D-25-42787R1

PLOS One

Dear Dr. Kabir,

I'm pleased to inform you that your manuscript has been deemed suitable for publication in PLOS One. Congratulations! Your manuscript is now being handed over to our production team.

Kind regards,

on behalf of

Dr. Xiaohui Zhang

Academic Editor

PLOS One